# FEATURE-LEVEL ADVERSARIAL ATTACK ON QUANTUM NEURAL NETWORKS

## ABSTRACT

Quantum Neural Networks (QNNs) have recently demonstrated promising performance in various tasks by leveraging the unique advantages of quantum computing. However, recent studies have also revealed the high sensitivity of QNNs to adversarial perturbations, posing a threat to their practical applications. Existing methods are developed under idealized assumptions, neglecting key practical constraints such as the inaccessibility of exact gradients and the stochasticity of quantum measurements in near-term noisy intermediate-scale quantum (NISQ) devices, thereby limiting their practical value. In this paper, we propose QMirage, a feature-level adversarial attack against QNNs that incorporates quantum-unique properties and resolves the gradient issue. Based on the definition of quantum latent features, we first introduce a new optimization objective to search for adversarial examples in feature space. We further employ natural evolution strategies (NES) with gradient priors for unbiased gradient estimation. Moreover, dynamic adjustment for the learning rate is combined to reduce failures caused by suboptimal fixed configurations. Experiments on benchmark datasets and QNNs demonstrate that QMirage achieves more effective and efficient attacks compared to baselines, while preserving comparable visual quality. It exhibits superior robustness under finite-shot and noisy settings with acceptable measurement costs. The results also reflect the influence of the model structure and encoding on adversarial robustness, providing insights for the future design of resilient QNNs.

## 1 INTRODUCTION

Over the past few years, the rapid development of quantum computing in both hardware and software has led to a novel research field, Quantum Machine Learning (QML) (Benedetti et al., 2019). On the one hand, machine learning has achieved initial success in quantum science (Carleo & Troyer, 2017; Zhang & Kim, 2017). On the other hand, quantum mechanisms and algorithms have also provided unique advantages to classical machine learning tasks (Cong & Duan, 2015; Maier et al., 2004; Rebentrost et al., 2013). Within QML, Quantum Neural Networks (QNNs) (Cong et al., 2018) have emerged as a recent outstanding branch, which integrates the intrinsic nature of quantum paradigms into neural networks, realizing time-complexity speedup (Cerezo et al., 2022) and model compression (Liu et al., 2025).

Adversarial attack (Goodfellow et al., 2014) is a well-documented challenge for deep neural networks (DNNs), where imperceptible perturbations are crafted to mislead models. Recently, to ensure the practical applications of QNNs along with advances in quantum hardware, concerns about adversarial vulnerabilities of QNNs (Lu et al., 2019) have become increasingly prominent. Beyond sensitivity to gradient-based perturbations, their robustness is influenced by inherent quantum characteristics such as system dimension (Liu & Wittek, 2019) and quantum noise (Cohen et al., 2019).

The most common adversarial attack methods for QNNs are adapted from classical gradient-based algorithms. These approaches generate either input-specific (Lu et al., 2019) or input-agnostic (Gong & Deng, 2022) perturbations, typically in an additive or functional manner by optimizing carefully crafted objective functions [1]. However, such methods heavily rely on gradient information,

---

[1]For clarity and consistency, the term objective function always denotes the final learning objective for a task, while the term loss is used to denote a single element in such a joint objective function.

which is often unavailable in realistic quantum settings because of certain encoding schemes and the constraints of actual quantum hardware. Also, the trade-off between success rate and visual quality remains a challenge in empirical evaluation. Recent efforts have shifted toward the use of quantum-specific properties, such as entanglement (Shi et al., 2025), to design attacks. Nonetheless, these methods often assume access to ideal quantum states, which are inherently inaccessible due to the irreversible collapse of quantum states on NISQ devices. These challenges highlight the urgent need to rethink the feasibility and effectiveness of adversarial attacks in practical quantum environments, where gradient access is limited and measurement is costly.

To address these issues, we propose a feature-level adversarial attack targeted at QNNs, named **QMirage**, which incorporates quantum-specific properties and addresses the challenge of inaccessible gradients in a real quantum context. Specifically, we define quantum latent features based on superposition and a new objective function that targets the disruption of latent features in QNNs, thereby accelerating the progress of the search for effective adversarial examples. To account for the unavailable gradients, we employ natural evolution strategies (NES) combined with time-dependent gradient priors to obtain unbiased gradients, making attacks applicable to various encoding schemes. In addition, a learning rate adjustment strategy is introduced to monitor the optimization trend and stabilize the generation of adversarial examples. Finally, the effectiveness of QMirage is validated on various benchmark datasets and QNN architectures, demonstrating its ability to generate more efficient and higher-quality adversarial examples compared to baseline methods. Results on finite-shot and noisy models further confirm its practical potential on future quantum devices.

Our contributions can be summarized as follows: (1) an objective function targeting superposition-based latent features in QNNs, (2) a comprehensive adversarial generation algorithm that considers both gradient estimation and learning rate adjustment, and (3) extensive experiments on various datasets and QNN structures whose results demonstrate the effectiveness of our attack.

## 2 RELATED WORK

Recent theoretical work (Liu & Wittek, 2019) has illustrated the vulnerability of QNNs to adversarial perturbations. Such vulnerability is related to several inherent quantum properties, including the dimension of quantum systems (Liu & Wittek, 2019), the randomness of the data distribution (Liao et al., 2021) and depolarizing noise (Cohen et al., 2019). Experimental validation has also witnessed a surge of related research (Lu et al., 2019; Ren et al., 2022).

**Classical-inspired Adversarial Attacks on QNNs.** Building on the similar optimization process of QNNs and DNNs, Lu et al. (2019) proposed quantum-adapted FGSM and BIM, which are input-specific and realized in both additive and unitary perturbations. Additive attacks remain consistent with their classical counterparts (i.e., adding perturbations directly to original inputs), whereas unitary attacks insert and optimize additional unitary operations to alter encoded quantum states. However, it is intractable to ensure a balance between attack success and quality in empirical implementation. They also lack the utilization of any quantum property of QNNs. Universal adversarial perturbations are also empirically verified in Gong & Deng (2022) by a classical generative network or an extra subcircuit. They suffer from low success rates and unavoidably disturb the model structures. On the other hand, current black-box attacks like the transfer attack (Lu et al., 2019; Wendlinger et al., 2024) focus mainly on the transferability between classical and quantum classifiers, whose success rate depends largely on model structures without a guarantee. The transferability between different quantum classifiers remains unexplored.

**Quantum-specific Adversarial Attacks on QNNs.** A more recent work, QuanTest (Shi et al., 2025) has made initial attempts to integrate quantum-unique properties into adversarial attacks. It designed the QEA criterion to quantify the entanglement change throughout the circuit and proposed a joint optimization problem. Their insight comes from the fact that inputs that produce higher QEAs should activate more entangled neurons and thus trigger misbehaviors. Despite the superiority in attack success and perturbation strength, its practical value is constrained by several factors. First, both the acquisition of native quantum states and the quantification of entanglement can only be achieved on ideal simulators. In real execution, the entanglement measure needs probability estimation through multiple measurements, and its accuracy is unavoidably affected by the inherent stochasticity. QuanTest also requires a copy of the input quantum state, which incurs additional resource costs and accumulates errors. Furthermore, its interpretability requires further improvement.

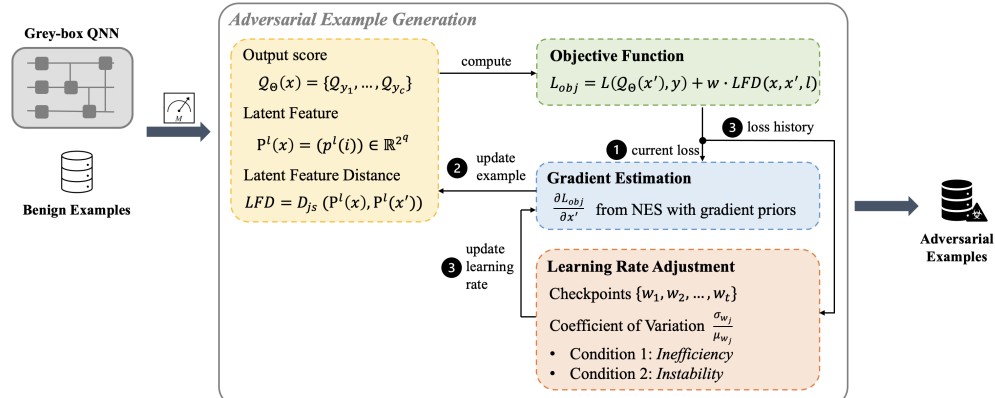

Figure 1: The overview of QMirage.

Lacking a rigorous definition of the entangled neuron in QNNs, the correlation between higher QEAs and increased model misbehaviors remains unclear.

More importantly, all gradient-based attacks fail to consider the accessibility of gradients in real-world quantum execution. Specifically, for QNNs that adopt amplitude encoding, the gradient with respect to the input cannot be obtained by the parameter shift rule, which is only applicable to gradients with respect to the model parameters. Also, constrained by the stochasticity of quantum measurement, automatic differentiation is no longer effective. For QuanTest, the gradient of the QEA with respect to inputs is essentially intractable.

**Adversarial defenses for QNNs.** Adversarial training (Lu et al., 2019) has been extended to the quantum setting by incorporating quantum adversarial examples into the training process, which improves the robustness of QNNs against adversarial attacks (Lu et al., 2020). Despite its early effectiveness, an important caveat is that defense is attack- and input-specific, resulting in poor generalization against fundamentally different attacks (Bai et al., 2021). In general, adversarial defenses for QNNs are still in their infancy and lack systematic methodologies.

## 3 THREAT MODEL

**The goal of the adversary.** With a focus on classification tasks, the adversary seeks to craft imperceptible and input-specific perturbations that are added to benign examples in order to mislead the target QNN. The perturbations are pixel-level, while leaving the circuit structure untouched to ensure stealthiness. Also, the adversary aims to improve the attack efficiency, considering the costs associated with quantum measurement.

**The knowledge of the adversary.** A *white-box* adversary is assumed to have complete access to the model and dataset, including model structure, parameters, gradients, and training data, while a *black-box* adversary is restricted to obtaining only the final outputs of the model (e.g., scores or labels) given limited queries. Our setting lies between these two scenarios, a *grey-box* adversary. Specifically, the adversary is assumed to have access to both model outputs and certain intermediate outputs (e.g., feature maps or activations) of the target QNN. However, they have no knowledge of the training data, the gradient information, the model parameters, or the detailed structure.

**The capabilities of the adversary.** The adversary is limited to introducing small imperceptible perturbations to existing inputs at inference time to construct adversarial examples. They cannot generate entirely synthetic inputs or arbitrarily modify the data distribution.

## 4 METHOD

Figure 1 illustrates the overview of QMirage. For each benign example, QMirage first obtains its original output scores of each class, the latent feature vector, and the latent feature distance (LFD, initialized as 0) (Section 4.1). During the generation phase, *objective function* is defined as the

maximization of the classification loss directed by LFD to induce the QNN misbehavior (Section 4.2). *Gradient estimation* employs natural evolution strategies (NES) combined with time-dependent gradient priors to obtain the gradient of the current loss, which serves as a feedback to update the example. Meanwhile, at each checkpoint, *learning rate adjustment* determines whether to update the learning rate based on loss history and two conditions of inefficiency and instability (Section 4.3). When the budget of benign examples is exhausted, a set of adversarial examples is produced.

Here, we first define several notations. A trained QNN with parameters $\Theta$ is denoted as $\mathcal{Q}_\Theta$. Since all quantum gates are unitary operations, we use $U_\Theta^l$ to represent the cumulative operations up to the $l$-th layer of a QNN, i.e., the production of unitaries before the $l$-th layer. For a $q$-qubit circuit, the quantum state after the $l$-th layer can be denoted as $|\phi^l\rangle = U_\Theta^l |0^{\otimes q}\rangle$. Dataset $\mathcal{T} = \{(x_1, y_1), ..., (x_n, y_n)\}$ contains $n$ benign inputs and their original labels for adversarial generation.

## 4.1 Objective Function

DNNs have been shown to be highly sensitive to perturbations that target inputs, neurons, or weights, creating vulnerabilities to adversarial attacks (Huang et al., 2023). To better understand such vulnerabilities, researchers have shifted attention to higher-level internal representations, which are referred to as feature maps extracted from an intermediate layer. They describe the collective patterns of neurons and the flow of feature transformation (Liao et al., 2018; Xie et al., 2019). Based on this perspective, feature-level attacks (Naseer et al., 2018b; Wang et al., 2021; Huang et al., 2023; Ganeshan et al., 2019) with a focus on internal feature distortion have achieved optimal attack success rates and transferability for DNNs. Inspired by that, we aim to validate the sensitivity of QNNs to such latent features. Given the absence of neurons and feature maps in QNNs, we first propose the definition of latent features for QNNs.

**Definition 4.1** (*Quantum Latent Feature*). The latent feature in the intermediate layer $l$ for the input $x$ is defined as the probability distribution $\mathbf{p}^l(x)$ of the corresponding quantum state. Specifically, $\mathbf{p}^l(x) = (p^l(i))_{i \in \{0,1\}^q} \in \mathbb{R}^{2^q}$, where $p^l(i) = |\langle i|\phi^l\rangle|^2$ is the probability of the quantum state $|\phi^l\rangle$ collapsing into the $i$-th computational basis state.

Probability distribution encodes the fundamental information of a quantum state, i.e., *superposition*, serving as an intuitive indicator of latent features in quantum circuits. However, other higher-level quantities, such as entanglement, are not directly observable and require additional procedures (Haug & Kim, 2023), causing increased resource overhead and inaccuracy in practice.

To activate abnormal intermediate model behaviors that interfere with the final prediction, we expect deviations between the latent features of the benign example and its adversarial version. We define the Latent Feature Distance (LFD) metric to quantify the degree of such a deviation:

$$LFD(x, x', l) = D_{js}(\mathbf{p}^l(x), \mathbf{p}^l(x')) \tag{1}$$

where $D_{js}$ refers to the Jensen-Shannon distance, a widely used metric to evaluate the difference between two probability distributions. It is calculated as $D_{js}(\mathbf{p}, \mathbf{q}) = \sqrt{\frac{1}{2}KL(\mathbf{p}||\mathbf{m}) + \frac{1}{2}KL(\mathbf{m}||\mathbf{q})}$ where $KL(\cdot||\cdot)$ denotes the Kullback-Leibler divergence and $\mathbf{m}$ is the average distribution of $\mathbf{p}$ and $\mathbf{q}$.

With fixed model parameters, an additive attack typically involves maximizing a specified objective loss to optimize over the input space, ultimately triggering abnormal outputs. Specifically, we adopt classification loss as our major attack objective and integrate LFD to guide the optimization direction, aiming for faster convergence towards the decision boundary. The objective function is defined as a joint optimization problem:

$$L_{obj} = L(\mathcal{Q}_\Theta(x_i'), y_i) + w \cdot LFD(x_i, x_i', l) \tag{2}$$

where $L(\mathcal{Q}_\Theta(x_i), y_i) = \max_{j \neq y_i} \mathcal{Q}_j(x_i) - \mathcal{Q}_{y_i}(x_i)$, $\mathcal{Q}_{y_i}(x_i)$ is the score for the input $x_i$ to be classified as $y_i$. Here, we focus on untargeted attacks. Targeted attacks can be easily implemented by adjusting the classification loss to focus on a specific target label. $w$ is the parameter to balance the strength between the two terms on optimization, whose value is determined by the relative scale between them.

## 4.2 GRADIENT ESTIMATION

Due to the special structures of quantum circuits, the gradient of a loss function with respect to the model parameters can be obtained directly through the parameter shift rule (Liu & Wang, 2018), depending on the pairwise evaluation at the shifted parameters. In the context of adversarial attacks, the optimization target transfers to a particular input example rather than the model parameters. For angle encoding, where input features are encoded as parameters of rotation gates, the parameter shift rule still works. However, for more complex encoding methods like amplitude encoding, where there exists no direct relation between input features and gate parameters, such gradients are not available in real quantum execution, falling into a black-box setting. To address this, we consider a zero-order method, natural evolution strategies (NES) (Wierstra et al., 2014), which only depends on function values to realize derivative-free optimization through a search distribution. In a manner similar to Ilyas et al. (2018a), we choose a search distribution of random Gaussian noise around the current input. And a two-query estimate and antithetic sampling are employed to obtain the gradient with respect to the input as follows:

$$\hat{g} := \frac{\partial f(x)}{\partial x} \approx \frac{1}{2\sigma K} \sum_{k=1}^{K} [(f(x + \sigma \cdot u_k) - f(x - \sigma \cdot u_k)) \cdot u_k] \tag{3}$$

where $\sigma$ is the search variance, $K$ is the sampling population, $u_k$ is sampled Gaussian vector, i.e., $u_k \sim \mathcal{N}(0, I)$. $f$ is any general function ($f = L_{obj}$ in our method).

As illustrated in Ilyas et al. (2018b), gradients exhibit time-dependent characteristics, which means that gradients of successive iterations are highly correlated and can serve as a predictor. We integrate it into our estimation to avoid excessive fluctuations in gradients across different iterations. Specifically, for the $t$-th iteration, after obtaining $\hat{g}^{(t)}$, it is further updated as $\hat{g}^{(t)} = (1-\beta) \cdot \hat{g}^{(t-1)} + \beta \cdot \hat{g}^{(t)}$, where $\beta$ regulates the influence of the previous update on the current gradient.

## 4.3 DYNAMIC ADJUSTMENT OF LEARNING RATE

The performance of existing additive attacks (Lu et al., 2019; Shi et al., 2025) for QNNs is further limited by a fixed learning rate (also called step size). This choice is suboptimal, since convergence is not guaranteed and attack performance can be highly influenced by a loss plateau or oscillation. To achieve a more robust attack, we utilize *checkpoints* (Croce & Hein, 2020) to monitor the optimization trend and adjust the learning rate, aiming to improve the general efficiency and stability. Note that the checking is based on the classification loss, considering that it directly correlates with attack success and indicates the current optimization trend.

Specifically, given a budget of $B$ iterations, checkpoints $w = \{w_0, w_1, \ldots, w_n\}$ are configured as $w_j = \lceil p_j B \rceil$ where internal $p_{j+1} = p_j + max\{p_j - p_{j-1} - 0.03, 0.06\}$ with $p_0 = 0, p_1 = 0.22$. As optimization proceeds, the check becomes gradually more frequent. During a checkpoint $w_j$, classification losses are denoted as $L_{w_j} = \{L_i, i = w_{j-1} + 1, \ldots, w_j\}$. To achieve dynamic adjustment, we first need to evaluate the variation of $L_{w_j}$. Variance is a standard statistical indicator of variability. However, it is insufficient as it can be influenced by the specific magnitude of the loss. Losses with larger magnitudes will have a larger absolute variance, even if they are less variable. Instead, we adopt the coefficient of variation (CV) (Groenendijk et al., 2021), which is calculated as the ratio of the standard deviation to the mean value for better quantification. This transforms different losses into a common scale, allowing for comparison independent of specific scales or magnitudes. Next, we consider two conditions for detecting inefficient or unstable optimization.

*Condition 1 (Inefficiency).* When loss plateaus or the learning rate is too small, optimization becomes inefficient without a significant improvement in loss. This finally consumes redundant iterations to find adversarial examples and can even become stuck with no progress. It is determined by:

$$\frac{\sigma_{w_j}}{\mu_{w_j}} < \tau_e \tag{4}$$

where $\mu_{w_j}$ and $\sigma_{w_j}$ denote the mean and standard deviation of $L_{w_j}$, and $\tau_e$ is the CV threshold for a minor variation and therefore inefficient optimization. If Condition 1 is true, the learning rate is increased twice to help escape from the plateau or accelerate the learning speed.

*Condition 2 (Instability).* Since the last checkpoint $w_{j-1}$, if the loss has not increased in most iterations and the variation is abnormally significant, the optimization is likely to exhibit oscillatory behavior. This condition is formulated as follows:

$$\sum_{i=w_{j-1}+1}^{w_j} \mathbf{1}_{L^{(i+1)}>L^{(i)}} < \rho \cdot (w_j - w_{j-1}) \text{ and } \frac{\sigma_{w_j}}{\mu_{w_j}} > \tau_s \quad (5)$$

where $L^{(i)}$ denotes the classification loss of $i$-th iteration, $\rho$ determines the minimum number of iterations in which the loss increased, and $\tau_s$ is the CV threshold for excessively large variation. If the first sub-condition is true, it implies that current optimization might not proceed stably as expected, during which at least a proportion of $(1 - \rho)$ iterations show fluctuations. If Condition 2 is true, the learning rate is halved to introduce a more moderate gradient updating.

Along with the adjustment, we restart from the historically optimal point that attains the highest classification loss, which can avoid a repeated search and achieve further acceleration.

### 4.4 ADVERSARIAL EXAMPLE GENERATION

---

**Algorithm 1:** Adversarial Example Generation

---

**Input:** Target QNN $\mathcal{Q}$, benign set $\mathcal{T}$, budget $B$, intermediate layer $l$, weight of LFD $w$
**Output:** Adversarial set $\mathcal{T}_a$

1 **def** GenerateAdversarialExamples()**:**
2      $\mathcal{T}_a \leftarrow \emptyset$
3      **for** $(x, y) \in \mathcal{T}$ **do**
4          $b, loss\_his, lr, g\_pre \leftarrow 0, \emptyset, 0.05, 0$
5          $(x', y') \leftarrow clone(x, y)$
6          **while** $b \leq B$ and $y' == y$ **do**
7              $L_{obj} \leftarrow L(\mathcal{Q}(x'), y) + w \cdot LFD(x, x', l)$
8              $loss\_his \leftarrow loss\_his \cup \{L\}$
9              $\hat{g} \leftarrow GradEst(L_{obj}, x', g\_pre)$
10              $x' \leftarrow clamp(x' + lr \cdot \hat{g})$
11              **if** $b$ is checkpoint **then**
12                  $lr, x_{best} \leftarrow AdjustLR(loss\_his)$
13                  $x' \leftarrow x_{best}$
14              $g\_pre \leftarrow \hat{g}$
15              $y' \leftarrow \mathcal{Q}.predict(x')$
16          **if** $y' \neq y$ **then**
17              $\mathcal{T}_a \leftarrow \mathcal{T}_a \cup \{x'\}$
18      **return** $\mathcal{T}_a$

---

Combined with the objective function, gradient estimation and dynamic adjustment, Algorithm 1 illustrates the complete generation process of QMirage. Given a QNN $\mathcal{Q}$, a budget of $B$ iterations, target intermediate layer $l$, the attack is carried out on each example $(x, y)$ in the benign set $\mathcal{T}$ to obtain the corresponding adversarial set (line 3). In each iteration, the algorithm computes the value of the objective function and updates the loss history with the current classification loss (lines 7-8). $x'$ is updated using estimated gradients with prior information (lines 9-10). When arriving at a checkpoint, the learning rate is dynamically adjusted based on the loss history using Conditions 1 and 2, and $x'$ is reset to the best one so far (lines 11-13). The current gradient is saved for the next iteration. If the current example $x'$ has become adversarial, it is appended to $\mathcal{T}_a$ (lines 16-17). The generation terminates until $\mathcal{T}$ is exhausted.

## 5 EXPERIMENT

### 5.1 SETTING

**Datasets and models.** As for benchmarks, we choose two widely adopted datasets for image classification tasks: MNIST and FashionMNIST. The MNIST dataset (LeCun et al., 1998) contains

70,000 grayscale images of handwritten digits (0–9), each with a resolution of $28 \times 28$. As a more challenging benchmark, the FashionMNIST dataset (Xiao et al., 2017) has the same format but comprises images from ten categories of clothing items. Regarding QNNs, we select four representative architectures with different circuit structures and encoding schemes, including QCL (Mitarai et al., 2018), QCNN (Cong et al., 2018), DRNN (Pérez-Salinas et al., 2020), and HQNN (Shi et al., 2024). Considering their scales and model accuracies, we implement binary classification on all models and ternary classification on QCL and QCNN. Details are listed in Table 4 in Appendix C.1.

**Evaluation metrics.** To evaluate the effectiveness of the attack, we adopt the Attack Success Rate (ASR), calculated as the proportion of adversarial examples generated within a given budget of iterations. In terms of attack efficiency, we consider the number of iterations needed to generate an effective adversarial example. For the example quality, the Structural Similarity Index Measure (SSIM) (Wang et al., 2004) is a perceptual metric that compares the similarity of two images, with values closer to 1 indicating greater attack quality. For iterations and SSIM, we report both their mean and standard variance values in the following experiments, presented as mean $\pm$ std.

**Parameter settings.** (1) To compute the LFD, we first need to specify the target intermediate layer. The choice varies for different QNNs according to their specific structures, as shown in Appendix C.1. (2) In the objective function, $w$ determines the strength of LFD loss and has a direct effect on the attack results. Based on the scale between classification loss and LFD, $w$ is set to 10 by default. Other settings will be explored in an ablation study (Appendix D.3). (3) For gradient estimation, $u_k$ is random Gaussian vector sampled from $\mathcal{N}(0, I)$. $\sigma$ is 0.1, and the sampling population for each estimation $K$ is 50. A larger $K$ brings a more accurate estimation at a higher time cost. $\beta$ is 0.9 for gradient priors. (4) For Condition 1 and 2, $\tau_e$, $\rho$ and $\tau_s$ are 0.01, 0.5 and 0.1. (5) For each example, the budget $B$ of iterations is chosen as 500.

**Baseline.** For comparison, we select several representative attacks as our baselines, which are input-specific and gradient-based like QMirage. Within classical-inspired attacks, qFGSM and qBIM (Lu et al., 2019) apply classical FGSM and BIM algorithms to QNNs and their classification loss function. We adapt optimization-based CW (Carlini & Wagner, 2017) for QNNs, which finds minimal perturbations that cause misclassification due to constrained loss. We also include QuanTest (Shi et al., 2025), which focuses on the quantum-unique entanglement property. To apply these attacks to amplitude-encoding QNNs under realistic conditions of estimated gradients, we also adopt NES for them with a same $K$ of 50. Detailed settings for baselines are listed in Appendix C.3.

We evaluate QMirage on QNNs using Pytorch 2.8 (Paszke et al., 2019) and PennyLane 0.42 (Bergholm et al., 2018). All experiments are conducted on systems equipped with Intel Xeon E5-1650 (6 cores, 32GB) and Ubuntu 22.04.

## 5.2 COMPARISON WITH BASELINE

### 5.2.1 ON IDEAL MODELS

Table 1: Gradient-targeted attack results on ideal QNNs (binary classification).

| | | QCL | | | QCNN | | | DRNN | | | HQNN | | |
|---|---|---|---|---|---|---|---|---|---|---|---|---|---|
| | | ASR | Iterations | SSIM | ASR | Iterations | SSIM | ASR | Iterations | SSIM | ASR | Iterations | SSIM |
| MNIST | qFGSM | 100% | / | 0.682±0.134 | 67.5% | / | 0.702±0.141 | 97.5% | / | 0.459±0.178 | 6% | / | 0.659±0.119 |
| | qBIM | 75% | 50±0 | 0.686±0.114 | 53.5% | 50±0 | 0.658±0.154 | 99% | 50±0 | 0.595±0.173 | 89.5% | 50±0 | 0.693±0.124 |
| | qCW | 4% | 30.75±41.48 | 0.894±0.078 | 10% | 95.84±76.41 | 0.729±0.115 | 43.5% | 28.74±78.53 | 0.794±0.138 | 4% | 113.88±127.29 | 0.988±0.07 |
| | QuanTest | 100% | 58.74±32.19 | 0.848±0.078 | 100% | 85.25±92.78 | 0.884±0.134 | 100% | 3.71±1.51 | 0.884±0.086 | 68% | 243.48±107.86 | 0.719±0.128 |
| | QMirage | 100% | 46.18±25.66 | 0.858±0.089 | 100% | 57.37±33.59 | 0.769±0.095 | 100% | 4.05±1.36 | 0.876±0.094 | 96% | 179.59±91.71 | 0.721±0.112 |
| FashionMNIST | qFGSM | 100% | / | 0.644±0.143 | 87% | / | 0.639±0.144 | 94% | / | 0.645±0.154 | 42% | / | 0.625±0.159 |
| | qBIM | 85.5% | 50±0 | 0.618±0.151 | 85.5% | 50±0 | 0.609±0.153 | 99% | 50±0 | 0.722±0.130 | 75% | 50±0 | 0.701±0.117 |
| | qCW | 13.5% | 10.82±11.71 | 0.901±0.067 | 44.5% | 42.41±63.35 | 0.815±0.099 | 96.5% | 5.72±26.74 | 0.881±0.065 | 42% | 91.88±125.46 | 0.917±0.121 |
| | QuanTest | 100% | 76.81±50.95 | 0.901±0.073 | 100% | 81.65±69.25 | 0.856±0.116 | 100% | 2.81±1.47 | 0.967±0.018 | 68.5% | 160.23±86.79 | 0.856±0.131 |
| | QMirage | 100% | 61.94±39.08 | 0.912±0.068 | 100% | 66.14±50.20 | 0.860±0.115 | 100% | 2.79±1.46 | 0.968±0.025 | 96.5% | 131.86±107.69 | 0.863±0.152 |

Considering that baselines were originally evaluated on ideal simulators with automatic differentiation, we first train QNNs under ideal settings, i.e., infinite-shot and noiseless simulator, and generate adversarial examples using baseline methods and QMirage, respectively. Results on binary- and ternary-classification QNNs are listed in Tables 1 and 6 (see Appendix D.1).

QMirage has achieved ASRs of 100% on almost all QNNs, indicating the high sensitivity of QNNs to feature-level distortions. Similar to the feature map in DNNs, latent features in QNNs can reflect rich feature extraction as the circuit deepens. These features are higher-level and more abstract, whose abnormal activations can induce errors in model predictions. A special case is HQNN, whose

ASR is only 96%. This is due to its special structure, where the first part is a classical linear layer for the feature extractor, and the second part is a small-scale quantum circuit as the final classifier. LFD targets the latent features specially extracted from the quantum part, with limited impact on the parameter-heavy classical part. Nevertheless, QMirage achieves optimal results compared to baselines. In terms of generation efficiency, QMirage requires fewer iterations to ensure attack success, thus reducing measurement overhead. It also performs well in visual quality, as reflected in high SSIMs. Despite the suboptimal SSIM on QCNN compared to QuanTest, QMirage is applicable in more realistic settings without measuring entanglement, as demonstrated in the following experiments. In a few cases, such as MNIST and QCL, qCW yields better SSIMs for its perturbation-constrained objective, yet this advantage is overshadowed by an extremely low ASR.

In addition, by comparing different QNNs, we can find that HQNN is more robust to adversarial perturbations, with the most iterations and the lowest SSIMs. As mentioned above, its hybrid structure makes the gradient update more complicated, making it more difficult to attack the two parts simultaneously. In contrast, DRNN is a more sensitive structure, for which QMirage needs only 4 iterations to mislead it on average. The sensitivity stems from the angle encoding, where the input features directly serve as gate parameters. The effect of perturbations accumulates along the circuit execution, resulting in a rapid movement toward the decision boundary.

In addition to these empirical attacks, we also compare QMirage with theoretically optimal adversarial examples, as solved by Guan et al. (2021), in Appendix D.2.

### 5.2.2 ON FINITE-SHOT AND NOISY MODELS

To explore adversarial attacks in real quantum execution, we train another family of QNNs in more realistic settings, that is, finite shots for obtaining outputs and the presence of quantum noise. Due to resource constraints, we turn to multiple shots and random noisy channels provided by Penny-Lane. Noisy channels include depolarizing, bit flip, phase flip, and phase damping. To alleviate computational overhead, these channels are approximated by inserting Pauli gates according to their definitions. Note that since QuanTest depends on the entanglement of native quantum states, which is inaccessible under finite shots, we exclude it here. As shown in Tables 2 and 7 (see Appendix D.1), compared to ideal settings, QMirage achieves a substantially greater advantage over baselines in all metrics. QMirage maintains especially higher ASRs and exhibits stronger robustness.

Table 2: Gradient-targeted attack results on finite-shot and noisy QNNs (binary classification).

|  |  | QCL | | | QCNN | | | DRNN | | | HQNN | | |
| --- | --- | --- | --- | --- | --- | --- | --- | --- | --- | --- | --- | --- | --- |
|  |  | ASR | Iterations | SSIM | ASR | Iterations | SSIM | ASR | Iterations | SSIM | ASR | Iterations | SSIM |
| MNIST | qFGSM | 39.5% | / | 0.653±0.148 | 44.5% | / | 0.653±0.148 | 91.5% | / | 0.469±0.198 | 2.5% | / | 0.660±0.117 |
|  | qBIM | 94.5% | 50±0 | 0.683±0.119 | 63% | 50±0 | 0.659±0.148 | 86.5% | 50±0 | 0.607±0.172 | 15.5% | 50±0 | 0.721±0.106 |
|  | qCW | 13.5% | 30.75±41.48 | 0.901±0.067 | 5.5% | 105.98±87.29 | 0.725±0.100 | 21.5% | 27.88±86.43 | 0.925±0.005 | 4.5% | 217.78±135.43 | 0.969±0.084 |
|  | QMirage | 100% | 42.06±26.29 | 0.845±0.076 | 100% | 50.53±35.98 | 0.742±0.093 | 100% | 4.05±1.36 | 0.831±0.100 | 91% | 179.59±91.71 | 0.687±0.146 |
| FashionMNIST | qFGSM | 41.5% | / | 0.611±0.153 | 39% | / | 0.651±0.142 | 92.5% | / | 0.651±0.154 | 38.5% | / | 0.612±0.161 |
|  | qBIM | 94.5% | 50±0 | 0.682±0.124 | 66.5% | 50±0 | 0.652±0.152 | 86% | 50±0 | 0.603±0.174 | 16.5% | 50±0 | 0.718±0.108 |
|  | qCW | 1.5% | 151.37±143.90 | 0.873±0.061 | 5.6% | 126.49±125.31 | 0.717±0.099 | 26.5% | 19.11±66.76 | 0.816±0.134 | 4% | 144.75±122.97 | 0.973±0.078 |
|  | QMirage | 100% | 56.01±32.67 | 0.883±0.091 | 100% | 60.76±47.65 | 0.840±0.112 | 100% | 3.57±2.09 | 0.933±0.050 | 95.5% | 92.87±11.06 | 0.747±0.192 |

By comparing a same QNN under ideal and realistic settings, all attacks show performance fluctuations. The first reason is the stochasticity of the measurement, which cannot be completely avoided despite multiple shots. The second is the estimated gradient. Its inaccuracy introduces loss fluctuations during optimization and requires stronger perturbations to cross the decision boundary. In particular, the unbiased estimation of NES indicates that, under infinite sampling, the objective gradient is equal to the expectation value of the loss function under a search distribution $\pi$, i.e., $\nabla_x \mathbb{E}_{\pi(\theta|x)}[F(\theta)] = \mathbb{E}[F(\theta)\nabla_x log\pi(\theta|x)]$. For a search distribution of random Gaussian noise where $\theta = x + \sigma u$ and $u \in \mathcal{N}$, estimating the gradient under sampling of $K$ times yields the variance-reduced gradient estimate as $\nabla \mathbb{E}[F_\sigma(x)] \approx \frac{1}{\sigma}\mathbb{E}[F(x + \sigma u)u] = \frac{1}{\sigma K}\sum_{i=1}^{K} u_i F(x + \sigma u_i)$. The practical estimation is influenced by $K$ and $\sigma$, i.e., more accurate with more sampling times and smaller noise $\sigma$. Furthermore, the estimated gradient $\hat{\nabla}$ with respect to the true gradient $\nabla$ can be bounded as (Ilyas et al., 2018a): $\mathbb{P}\{(1-\delta)||\nabla||^2 \leq ||\hat{\nabla}||^2 \leq (1+\delta)||\nabla||^2\} \geq 1 - 2p$ where $0 < \delta < 1$ and $n = O(-\delta^{-2}log(p))$, indicating more sampling costs for more accurate estimation and stable attack performance.

This randomness and inaccuracy involved in real quantum execution should not be ignored, since they unavoidably affect the evaluation. Hence, when designing testing or verification techniques tailored for QNNs in the future, such a simulation is necessary to avoid overestimation.

### 5.3 ABLATION STUDY: THE EFFECTIVENESS OF LFD

The core component of QMirage is the LFD loss term. Here we validate its guidance effect under two settings: without and with LFD in the objective function, configured as $w = 0$ and $w = 10$ respectively.

Table 3: Attack performance of QMirage without and with LFD guidance (MNIST).

| | QCL | | QCNN | | DRNN | | HQNN | | QCL-3 | | QCNN-3 | |
|---|---|---|---|---|---|---|---|---|---|---|---|---|
| | w/o | w/ | w/o | w/ | w/o | w/ | w/o | w/ | w/o | w/ | w/o | w/ |
| ASR | 99% | 100% | 91% | 100% | 100% | 100% | 89% | 91% | 100% | 100% | 100% | 100% |
| Iterations | 61.41±48.67 | 42.06±26.29 | 75.21±61.27 | 50.53±35.98 | 4.93±1.99 | 4.74±1.90 | 240.51±89.64 | 187.13±115.25 | 27.70±19.37 | 24.66±18.85 | 31.55±32.03 | 29.35±31.14 |
| SSIM | 0.895±0.064 | 0.845±0.076 | 0.816±0.073 | 0.742±0.093 | 0.869±0.083 | 0.831±0.100 | 0.679±0.152 | 0.678±0.146 | 0.957±0.052 | 0.934±0.086 | 0.949±0.070 | 0.921±0.133 |

As shown in Table 3, with the guidance of LFD, generation requires fewer iterations, resulting in a maximum improvement of 32.81% (QCNN). Also, ASRs of partial QNNs are increased, implying the role of LFD in guiding the optimization towards the decision boundary. However, compared with the objective without LFD, the one with LFD guidance may introduce additional perturbations and result in lower SSIMs in general. Considering the success of the attack and measurement costs, the slight decrease in visual quality is still acceptable. On DRNN, QCL-3, and QCNN-3, the acceleration is not obvious because these models are more sensitive to perturbations and yield wrong predictions in a few iterations. The guidance effects of LFD are more obvious with a larger $w$, as discussed in Appendix D.3.

### 5.4 DISCUSSION: MEASUREMENT COSTS

In the context of QNN, the need for more iterations to generate an adversarial example causes not only higher time costs but also increased measurement-related resource consumption. In addition to the measurements required for obtaining circuit outputs and estimating gradients, which are also involved in baseline methods, the calculation of LFD in QMirage introduces additional measurements. Here, we assess the measurement costs of QMirage to examine its practical feasibility.

To obtain QNN outputs, consider $N$ shots and $n_q$ qubits to produce outputs (e.g., 2 for binary classification). Since the output score for a specific class is treated as the expectation value of the corresponding qubit, the total measurement cost of obtaining outputs is $n_q N$. A latent feature is composed of probability amplitudes of all basis states, which can be approximated simultaneously within $N$ shots. Hence, for each input, obtaining its original output scores and latent feature vector requires $(n_q + 1)N$ shots in total before the generation loop. In each iteration, the first source of measurement also comes from extracting current output and latent features, that is, $(n_q + 1)N$. Another source is sampling for gradient estimation. According to Equation 3, the objective function $L_{obj}$ needs to be executed twice in each sampling, causing $2(n_q + 1)N$ shots. Hence, the cost for one iteration is $(2K + 1)(n_q + 1)N$, where $K$ represents the sampling population size. Given a budget of $B$ iterations for generation, QMirage consumes $[(2K + 1)B + 1](n_q + 1)N$ shots in total.

We observe that the maximum measurement for generating an adversarial example scales linearly with the number of output qubits and the sampling times for gradient estimation. In the future, more precise and efficient gradient estimation methods could be developed to improve their impact on physical resources.

## 6 CONCLUSION

In this paper, we proposed QMirage, a feature-level adversarial attack tailored for QNNs. By introducing quantum latent features based on superposition, a constrained objective function is optimized to trigger model misbehaviors. For improved feasibility in practice, QMirage integrates gradient estimation with prior information to accommodate various encoding schemes. It also addresses the suboptimal choice of the manual learning rate using a dynamic adjustment mechanism. Extensive experiments on benchmark QNNs and datasets demonstrated the effectiveness, robustness, and feasibility of QMirage under both ideal and realistically simulated settings. The results further highlight the need to consider additional factors involved in real quantum hardware to avoid overestimation. Future directions include exploring phase-sensitive latent features and developing defense mechanisms against these feature-level attacks.

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

# A    BASIC KNOWLEDGE OF QUANTUM COMPUTING

| Gate | Symbol | Matrix |
|---|---|---|
| Pauli-X | $-\boxed{X}-$ | $\begin{bmatrix} 0 & 1 \\ 1 & 0 \end{bmatrix}$ |
| Hadamard | $-\boxed{H}-$ | $\frac{1}{\sqrt{2}}\begin{bmatrix} 1 & 1 \\ 1 & -1 \end{bmatrix}$ |
| Rotation-X | $-\boxed{R_x(\theta)}-$ | $\begin{bmatrix} \cos(\theta/2) & -i\sin(\theta/2) \\ -i\sin(\theta/2) & \cos(\theta/2) \end{bmatrix}$ |
| CNOT | | $\begin{bmatrix} 1 & 0 & 0 & 0 \\ 0 & 1 & 0 & 0 \\ 0 & 0 & 0 & 1 \\ 0 & 0 & 1 & 0 \end{bmatrix}$ |

Figure 2: Common quantum gates

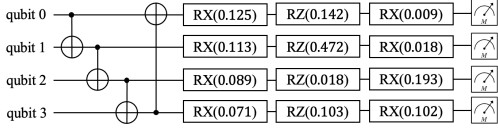

Figure 3: Example of a parameterized quantum circuit.

**Qubit.** A quantum bit, or qubit, is the fundamental unit of information in quantum computing. Unlike classical bits, qubits can exist in a *superposition* of computational basis states. A pure state is written as $|\phi\rangle = \alpha|0\rangle + \beta|1\rangle$ with $\alpha, \beta \in \mathbb{C}$ as *probability amplitudes*. Upon measurement, the qubit collapses to $|0\rangle$ with probability $|\alpha|^2$ and to $|1\rangle$ with probability $|\beta|^2$. In polar form, $\alpha, \beta$ can be mapped onto the Bloch sphere for geometric representation.

**Quantum gate and circuit.** Quantum gates are essential components in quantum programs that perform rotation and entanglement on qubits. Some common quantum gates are depicted in Figure 2. Quantum circuits are compositions of qubits and quantum gates, as shown in Figure 3. Quantum circuits realize various functionalities by modifying the overall structure, which is reflected in the selection and parameterization of quantum gates, the choice of target qubits, and the specific execution order.

**Quantum measurement.** Quantum measurement projects a superposition into a definite state according to probability amplitudes, with the state collapsing irreversibly. This operation is conventionally performed at the terminal stage of a quantum circuit. Due to the probabilistic nature, the output of a quantum circuit needs multiple measurements to achieve a reliable and comprehensive evaluation.

**Parameterized quantum circuit.** A Parameterized Quantum Circuit (PQC) consists of a fixed circuit structure with trainable gate parameters, making it suitable for optimization problems, especially in the context of machine learning tasks. Parameterized gates can be treated either as learnable components that are tuned to optimize the loss or as data-encoding blocks that map classical data into quantum states. By measuring qubits, we extract classical information from non-deterministic quantum paradigms, which can be applied to various downstream tasks.

**Quantum gradient estimation.** *Parameter shift rule* (Liu & Wang, 2018) is widely used to compute gradients in parameterized quantum circuits. Unlike backpropagation in classical neural networks, quantum circuits suffer from the irreversible measurement process and inherent noise, making automatic differentiation infeasible and necessitating specialized gradient estimators. According to the parameter shift rule, the derivative of an expectation value $E$ with respect to a gate parameter $\theta$ can be expressed exactly as a linear combination of expectation values evaluated at values of shifted parameters, such as $\frac{\pi}{2}$. Specifically, it is defined as:

$$\frac{\partial E(\Theta)_\theta}{\partial \theta} = \frac{1}{2}(E(\Theta)_{\theta + \frac{\pi}{2}} - E(\Theta)_{\theta - \frac{\pi}{2}}) \tag{6}$$

where $E$ is the overall operation including unitary gates and measurement operators. Compared to finite-difference methods, the parameter shift rule is unbiased and able to provide exact gradient estimation (Lu et al., 2019). Since it depends only on the measurement results, it can be applied directly to quantum hardware in practice, while backpropagation is only feasible on ideal simulators. Although this rule supports the gradients of $E$ with respect to gate parameters, it cannot differentiate with respect to given inputs, making it ineffective in producing gradient-based adversarial examples.

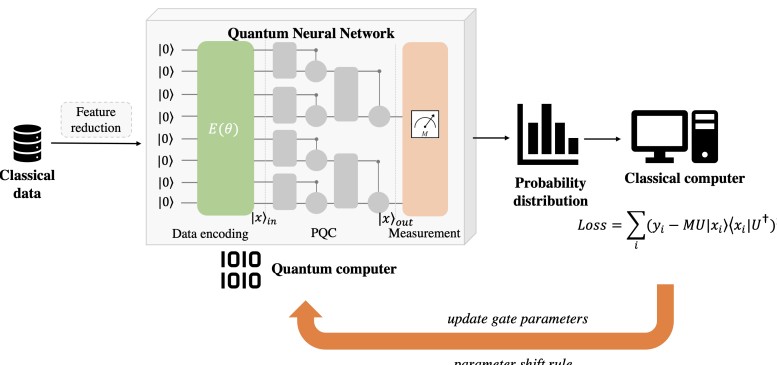

Figure 4: Three components of QNNs, including data encoding layer, parameterized circuit layer, and measurement layer.

## B ADDITIONAL RELATED WORK

### B.1 QUANTUM NEURAL NETWORKS

Inspired by DNNs, QNNs have become a core component of QML, targeting optimization tasks. Instead of neurons, QNNs are typically constructed from PQCs with fixed circuit structure and tunable gate parameters. Figure 3 shows a simple example of a PQC. It consists of a linear entangling layer with CNOT gates between neighboring qubits, as well as parameterized single-qubit rotation gates. Each rotation angle of an $R_x$ or $R_z$ gate is trainable during the optimization process. In Figure 4, a QNN typically comprises three main components: a data encoding layer to encode classical data into quantum states, a PQC layer to transform states, and a measurement layer to extract classical information from quantum circuits. The optimization computation and parameter update are conducted on classical computers.

Various QNN variants have been proposed (Shi et al., 2024; Cong et al., 2018; Mitarai et al., 2018; Hur et al., 2021). Based on circuit functionality, current QNNs can be categorized into three types: *circuit-body QNNs* Mitarai et al. (2018); Cong et al. (2018), which use medium-sized PQCs as the backbone; *circuit-kernel QNNs* Henderson et al. (2020), which employ PQCs as convolutional kernels for feature extraction; and *hybrid QNNs* Shi et al. (2024), which integrate PQCs with classical layers, where PQCs serve as either a preprocessing or output layer.

For the data encoding layer, two approaches are commonly used. *Amplitude encoding* encodes data features as amplitudes of a quantum state, which require relatively few qubits but deep circuits to implement. *Angle encoding* encodes features as rotation-gate parameters, enabling efficient implementation but consuming more qubits. For a parameterized circuit layer, two representative designs are *block stacking* (Mitarai et al., 2018), which repeatedly applies the same blocks, and *hierarchical structures* (Cong et al., 2018), which reduce circuit freedom by measuring subsets of qubits during execution.

### B.2 ADVERSARIAL ATTACKS FOR DNNS

• **White-box attack.** These methods typically assume that the attacker has full knowledge of the target model, including its architecture, parameters, and gradients. Early studies focused on deep neural networks (DNNs) under this setting. FGSM (Goodfellow et al., 2014) first revealed the vulnerability of image classifiers by perturbing inputs in the direction of the gradient in a single step. CW (Carlini & Wagner, 2017) and PGD (Madry et al., 2017) extended this idea with optimization-based and iterative strategies, respectively. DeepFool (Moosavi-Dezfooli et al., 2016) estimates the nearest decision boundary and generates minimal perturbations to cross it. Most subsequent white-box attack methods (Gao et al., 2020; Dong et al., 2023; Gu et al., 2022) can be regarded as extensions or refinements of these foundational approaches. These methods laid the foundation for evaluating robustness in DNNs.

• **Black-box attack.** These methods assume that the adversary has no access to the parameters or gradients of the target model and must rely on model queries or the transferability of adversarial

examples. A common approach is transfer-based attacks, where adversarial examples are crafted on surrogate models and then transferred to the target (Papernot et al., 2017; Chen et al., 2023). Another line of work is score-based attacks, which leverage confidence scores returned by the model to estimate gradients, such as ZOO (Chen et al., 2017) and NES (Ilyas et al., 2018a). In contrast, decision-based attacks operate under the label-only setting, relying solely on predicted labels to guide perturbations, with Boundary Attack (Brendel et al., 2017) and HopSkipJump (HSJA) (Chen et al., 2020) as representative methods. These three categories form the foundation of black-box adversarial attack research, with later extensions targeting more complex models.

• **Attack based on latent features.** Intermediate feature maps encode high-level, task-relevant semantics, motivating the development of latent feature-based adversarial attacks. For example, Feature Disruption Attack (FDA) (Ganeshan et al., 2019) introduces perturbations that corrupt deep features at multiple network layers, significantly degrading model performance. The Neural Representation Distortion Method (NRDM) (Naseer et al., 2018a) exploits perceptual similarity metrics and generalizability of neural features, resulting in effective untargeted attacks with strong cross-model and cross-task transferability. Meanwhile, Feature Importance-aware Attack (FIA) (Wang et al., 2021) specifically targets object-aware features that consistently dominate model decisions, further improving transferability across different architectures. Collectively, these works confirm that feature maps play a pivotal role in the generation of adversarial examples, offering both theoretical and practical guidance for the development of our approach.

## C EXPERIMENTAL DETAILS

### C.1 DATASETS AND QNN ARCHITECTURES

Table 4: Dataset and QNN architectures

| Dataset | Task | Target classes | QNN | #gate | Ideal Acc (%) | Realistic Acc (%) |
|---|---|---|---|---|---|---|
| MNIST | Binary classification | digits 0 and 1 | QCL | 150 | 100 | 100 |
| | | | QCNN | 180 | 100 | 100 |
| | | | DRNN | 120 | 99.29 | 99.53 |
| | | | HQNN | 4168 | 100 | 100 |
| | Ternary classification | digits 0, 1 and 2 | QCL | 150 | 91.86 | 91.06 |
| | | digits 4, 5 and 7 | QCNN | 180 | 90.16 | 87.56 |
| FashionMNIST | Binary classification | T-shirt and Trouser | QCL | 150 | 92.5 | 91.25 |
| | | | QCNN | 180 | 93 | 91.75 |
| | | | DRNN | 120 | 92.21 | 94 |
| | | | HQNN | 4168 | 97 | 97 |
| | Ternary classification | T-shirt, Trouser and Pullover | QCL | 150 | 89.67 | 88.67 |
| | | Trouser, Pullover and Dress | QCNN | 180 | 91.5 | 90.5 |

Here, we introduce the details of the circuit structures and encoding methods for each QNN in our experiments.

**Quantum Circuit Learning (QCL)** (Mitarai et al., 2018) belongs to block-stacking structure and adopts amplitude encoding, as shown in Figure 5(a). Each block contains three single-qubit rotation gates ($R_x, R_z, R_x$) and nearest-neighbor entangling gates (CNOTs). The circuit stacks five identical layers, with a target layer of 4 when extracting latent features.

**Quantum Convolutional Neural Network (QCNN)** (Cong et al., 2018) follows a hierarchical structure with amplitude encoding. Inspired by CNNs, it comprises Convolutional 1, Pooling 1, Convolutional 2, Pooling 2, and a Fully Connected layer as shown in Figure 5(b). Each convolutional block applies single-qubit rotation gates ($U_3, R_y, R_z$) and CNOTs to transform features, while the pooling blocks progressively reduce the number of qubits via controlled gates. The final results are read after fully-connected layer. For clarity, we group QCNN into three layers: $l_1 = \{conv1, pool1\}$, $l_2 = \{conv2, pool2\}$, and $l_3 = \{FC\}$. The target layer is 2, i.e., before the FC layer.

**Data Re-uploading Neural Network (DRNN)** (Pérez-Salinas et al., 2020) adopts a block-stacking ansatz with angle encoding. The quantum state is initialized as $|0^{\otimes q}\rangle$. In Figure 5(c), each layer contains three components: an encoding layer to embed input features as rotation angles for three rotation gates, a trainable layer to apply learnable $R_x$, and an entangling layer of nearest-neighbour CRZ gates. The total number of layers is 4, and the target layer is set to 3.

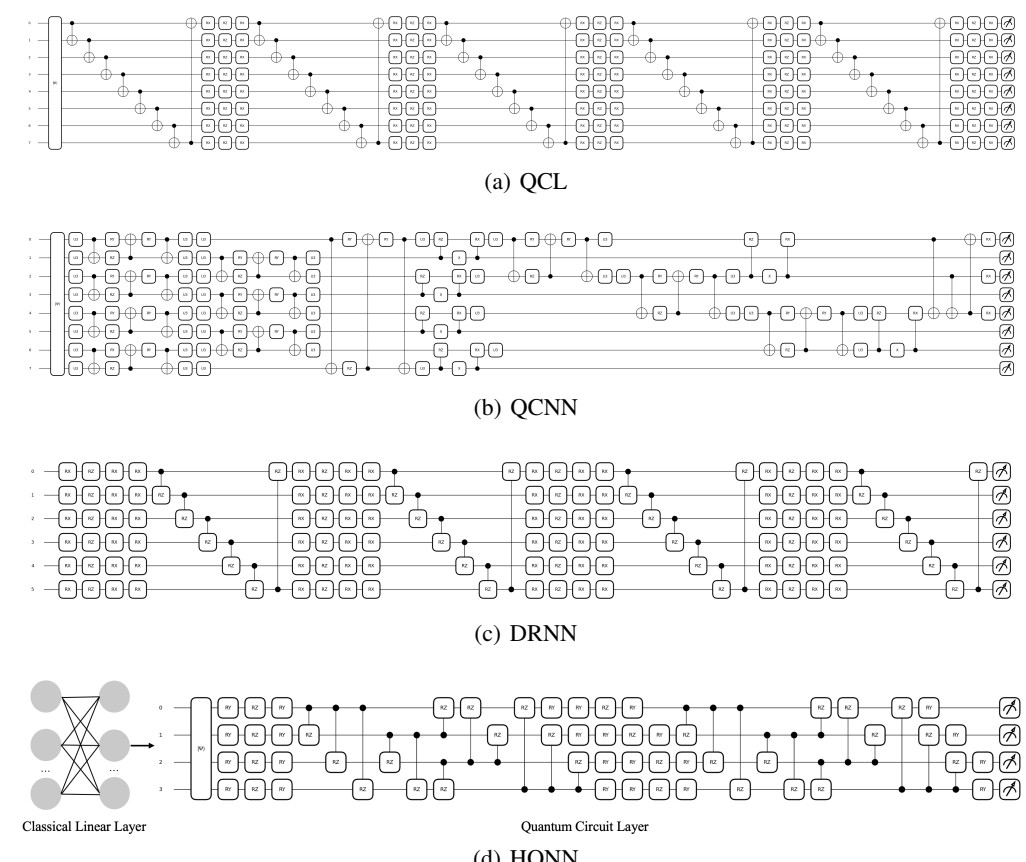

(a) QCL

(b) QCNN

(c) DRNN

(d) HQNN

Figure 5: 8-qubit QNN structures in our experiments.

**Hybrid Quantum Neural Network (HQNN)** (Shi et al., 2024) consists of a classical linear layer and a 4-qubit quantum circuit as in Figure 5(d). The classical part downsamples the original inputs to those whose shapes are $2^4$. The quantum part is block-stacking whose each block contains two rotation layers and one entangling layer. It stacks two blocks, and the target layer is 1.

Considering the time overhead, we adjust the image size of the DRNN to $8 \times 8$ and configure it with 6 qubits, while the other QNNs are set to 8 qubits and the images are downsampled to $16 \times 16$.

## C.2 MODEL TRAINING

Considering the current QNN scale, we adopt 20% of the original training and test data following Hur et al. (2021). We trained the QNN for 20 epochs with a batch size of 64 using the Adam optimizer. Adam optimizer was used with an initial learning rate of 0.01, $\beta_1 = 0.9$, $\beta_2 = 0.999$, $\epsilon$ = 1e-8, and weight decay=0. The learning rate decayed at epochs 5, 10, 15, and 20 following a MultiStepLR schedule. The gradient computation of both ideal and finite-shot models is based on parameter shift rule in PennyLane.

## C.3 CONFIGURATIONS OF BASELINE ATTACKS

Table 5: Configurations for baseline attacks.

| Attack | Configuration |
|---|---|
| qFGSM | $\epsilon = 64/255$ |
| qBIM | $\epsilon = 64/255, \alpha = 12/255, steps = 50$ |
| qCW | $L_2$ constraint, $c = 30, steps = 500, lr = 0.01$ |
| QuanTest | $QEA\_k = 1, QEA\_w = 1, lr = 0.05$ |

Here we give the detailed configurations for each baseline attack in Table 5. $\epsilon$ denotes the maximum perturbation applied to the examples, $\alpha$ denotes the step size, and $c$ denotes the weight of the classification loss in the objective function of CW. For QuanTest, $QEA\_k$ is the weight in Equation (7) of the original paper, and $QEA\_w$ is the weight of QEA in the objective function.

# D  ADDITIONAL EXPERIMENTS

## D.1  MORE RESULTS OF COMPARISON WITH BASELINE

The attack results on ideal and realistic QNNs of ternary classification are listed in Tables 6 and 7.

Table 6: Gradient-targeted attack results on ideal QNNs (ternary classification).

|  |  | QCL-3 | | | QCNN-3 | | |
|---|---|---|---|---|---|---|---|
|  |  | ASR | Iterations | SSIM | ASR | Iterations | SSIM |
| MNIST | qFGSM | 100% | / | 0.723±0.121 | 94.67% | / | 0.749±0.060 |
|  | qBIM | 90.33% | 50±0 | 0.729±0.112 | 71.33% | 50±0 | 0.776±0.053 |
|  | qCW | 76.67% | 41.82±63.89 | 0.918±0.081 | 38.33% | 49.32±91.54 | 0.934±0.063 |
|  | QuanTest | 100% | 38.02±26.67 | 0.927±0.082 | 100% | 25.84±13.27 | 0.946±0.134 |
|  | QMirage | 100% | 31.38±20.65 | 0.942±0.073 | 100% | 21.08±14.72 | 0.950±0.047 |
| FashionMNIST | qFGSM | 97.33% | / | 0.589±0.162 | 71% | / | 0.567±0.136 |
|  | qBIM | 82% | 50±0 | 0.587±0.171 | 62.33% | 50±0 | 0.594±0.127 |
|  | qCW | 84.67% | 38.06±64.46 | 0.842±0.107 | 49.67% | 13.25±37.72 | 0.838±0.132 |
|  | QuanTest | 100% | 74.31±63.44 | 0.889±0.096 | 100% | 80.24±88.18 | 0.806±0.175 |
|  | QMirage | 100% | 65.12±47.46 | 0.913±0.089 | 100% | 78.02±80.19 | 0.821±0.175 |

Table 7: Gradient-targeted attack results on finite-shot and noisy QNNs (ternary classification).

|  |  | QCL-3 | | | QCNN-3 | | |
|---|---|---|---|---|---|---|---|
|  |  | ASR | Iterations | SSIM | ASR | Iterations | SSIM |
| MNIST | qFGSM | 70.33% | / | 0.679±0.114 | 62.33% | / | 0.687±0.078 |
|  | qBIM | 100% | 50±0 | 0.714±0.116 | 98.67% | 50±0 | 0.730±0.066 |
|  | qCW | 36.33% | 17.36±29.94 | 0.918±0.059 | 17.67% | 54.34±86.99 | 0.932±0.053 |
|  | QMirage | 100% | 24.66±18.85 | 0.934±0.086 | 100% | 29.35±31.14 | 0.921±0.133 |
| FashionMNIST | qFGSM | 54.67% | / | 0.565±0.164 | 48% | / | 0.613±0.176 |
|  | qBIM | 97% | 50±0 | 0.595±0.151 | 85.67% | 50±0 | 0.629±0.168 |
|  | qCW | 33% | 45.56±72.85 | 0.812±0.115 | 33% | 30.75±72.85 | 0.848±0.149 |
|  | QMirage | 100% | 48.96±47.10 | 0.901±0.126 | 100% | 52.51±52.76 | 0.866±0.179 |

## D.2  COMPARISON WITH OPTIMAL ROBUSTNESS BOUND

Table 8: Comparison between RobustnessVerifier and QMirage

|  | QCL | | | QCNN | | | DRNN | | | HQNN | | | QCL-3 | | | QCNN-3 | | |
|---|---|---|---|---|---|---|---|---|---|---|---|---|---|---|---|---|---|---|
|  | ASR | Time (s) | Fidelity | ASR | Time (s) | Fidelity | ASR | Time (s) | Fidelity | ASR | Time (s) | Fidelity | ASR | Time (s) | Fidelity | ASR | Time (s) | Fidelity |
| RobustnessVerifier | 100% | 51.35 | 0.932 | 100% | 57.48 | 0.861 | N/A | N/A | N/A | 43.5% | 1.19 | 0.003 | 100% | 55.07 | 0.983 | 100% | 55.61 | 0.969 |
| QMirage | 100% | 12.65 | 0.901 | 100% | 17.24 | 0.807 | 100% | 1.11 | 0.999 | 96% | 19.42 | 0.079 | 100% | 9.18 | 0.971 | 100% | 8.88 | 0.959 |

RobustnessVerifier (Guan et al., 2021) proposed a robustness verification algorithm formulated in the quantum matrix form, which derives both the optimal robustness bound and adversarial examples for quantum classifiers. Based on the basic postulate of linearity in quantum mechanics, the algorithm is represented as a Constraint Satisfaction Problem (CSP), which is solved by calling a Quadratically Constrained Quadratic Program (QCQP) solver for the image classification task, where images are encoded in non-convex pure states. Let $U_\Theta$ be the matrix form of the parameterized layer of the QNN, $\{M_k\}_{k \in C}$ be the measurement operators corresponding to different classes, $|\phi\rangle$ be the initial quantum state encoded by an input with $\mathcal{Q}(|\phi\rangle\langle\phi|) = l$ where $l$ is the original label. $\delta$ is the optimal robust bound for $|\phi\rangle$ against adversarial perturbations, where $\delta = min_{k \neq l}\delta_k$ and $\delta_k$ is the solution for the problem as:

$$\delta_k = \min_{|\phi\rangle \in \mathcal{H}} \quad 1 - \langle\varphi|\phi\rangle\langle\phi|\varphi\rangle$$
$$\text{subject to} \quad \langle\varphi|\varphi\rangle = 1$$
$$\langle\varphi|U_\Theta^\dagger(M_l^\dagger M_l - M_k^\dagger M_k)|\varphi\rangle \leq 0$$

To further validate the effectiveness of QMirage, we investigate the gap between theoretically optimal adversarial examples and those generated by QMirage. We implement RobustnessVerifier by adapting the public code [2] to our QNNs. Considering that CSP is defined in terms of the matrix formulation of quantum circuits and states, QMirage is configured to ideal settings, as described in

---

[2]https://github.com/Veri-Q/Robustness

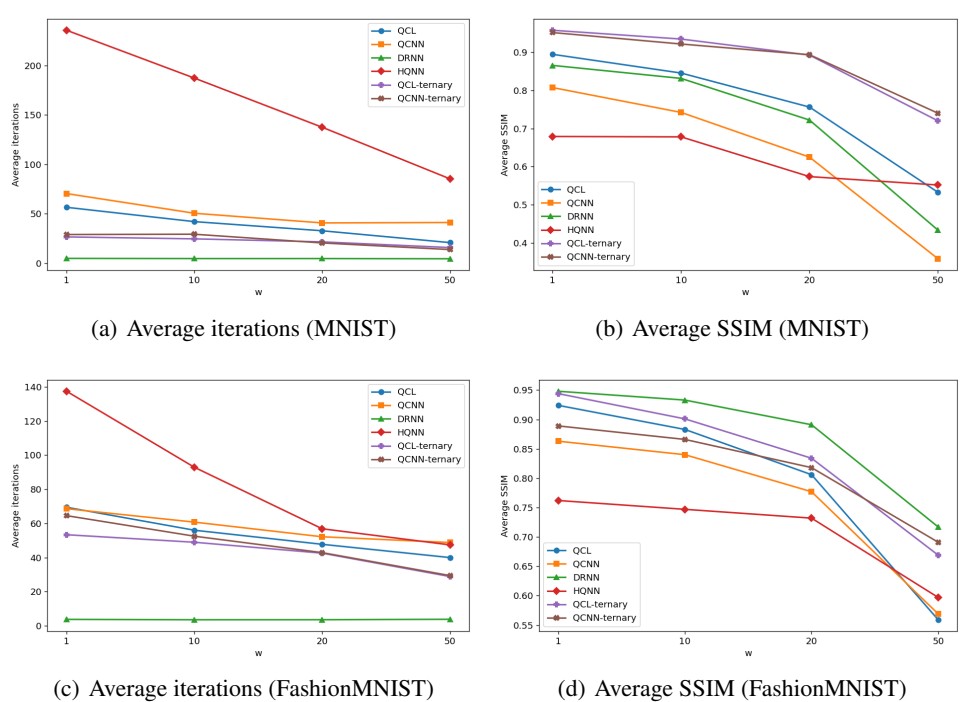

Figure 6: Effect of weight of LFD on attack performance of QMirage.

Section 5.2.1. With respect to metrics, we adopt fidelity as an alternative to SSIM, which is one of the most common quantities for measuring the similarity between two quantum states. The results of ASR, the average generation time, and the fidelity are shown in Table 8.

We observe that the example quality of QMirage is highly comparable to that of RobustnessVerifier with superior efficiency, especially on ternary-classification models. This further demonstrates the efficiency and effectiveness of QMirage. Also, limited by the separate representation of the initial quantum state and the following parameterized circuit, RobustnessVerifier can only deal with circuit-body QNNs using amplitude encoding. Angle-encoding variants, such as DRNN, which encode input features individually at different depths of the circuit, are not effectively supported. A special case is the fidelity on HQNN, 0.003 and 0.079, produced by two methods, respectively. This stems from the fact that the classical linear layer of HQNN extracts initial features from perturbed inputs, amplifying the difference between original and perturbed examples in feature space before the quantum part. This might provide insights for future defense strategies against adversarial attacks.

## D.3 ABLATION STUDY: WEIGHT OF LFD

In Section 5.3, we demonstrated the effective guidance of LFD loss to improve generation efficiency. However, convergence and example quality are affected by the specific choice of $w$, i.e., a larger $w$ introduces more distortions and a smaller one impairs the guidance of LFD. To figure out this effect on QMirage, we configure $w$ as 1, 10 (default setting), 20, and 50, respectively, and report results in Figure 6.

As expected, a larger $w$ brings about fewer iterations and lower SSIMs. It accelerates the convergence while sacrificing the imperceptibility of perturbations. However, an aggressive $w$ value, such as over 50, might have negative effects, causing optimization to focus too much on increasing the LFD, resulting in sharper fluctuations. The general magnitude of the gradients also increases accordingly, causing stronger perturbations. We will consider a more dynamic selection of $w$ in future work to alleviate the suboptimal effects of manual choice.

Table 9: Results of attacking different target layers in QNNs. $l_*$ denotes $*$-th layer for a particular QNN.

| | | QCL | | | QCNN | | DRNN | | | QCL-3 | | | QCNN-3 | |
|---|---|---|---|---|---|---|---|---|---|---|---|---|---|---|
| | | $l_2$ | $l_3$ | $l_4$ | $l_1$ | $l_2$ | $l_1$ | $l_2$ | $l_3$ | $l_2$ | $l_3$ | $l_4$ | $l_1$ | $l_2$ |
| MNIST | ASR | 100% | 100% | 100% | 100% | 100% | 100% | 100% | 100% | 100% | 100% | 100% | 100% | 100% |
| | Iterations | 43.88±26.39 | 43.66±26.45 | 42.06±26.29 | 52.04±37.67 | 50.53±35.98 | 4.91±1.93 | 4.78±1.81 | 4.80±1.99 | 29.59±49.92 | 28.57±46.16 | 24.66±18.85 | 25.99±26.23 | 29.35±31.13 |
| | SSIM | 0.842±0.075 | 0.844±0.074 | 0.845±0.076 | 0.733±0.099 | 0.742±0.093 | 0.858±0.092 | 0.835±0.097 | 0.826±0.101 | 0.923±0.133 | 0.925±0.119 | 0.934±0.086 | 0.931±0.075 | 0.921±0.133 |
| FashionMNIST | ASR | 100% | 100% | 100% | 100% | 100% | 100% | 100% | 100% | 100% | 100% | 100% | 100% | 100% |
| | Iterations | 58.81±34.95 | 59.22±34.88 | 54.45±30.97 | 60.66±43.53 | 60.75±47.65 | 3.74±2.16 | 3.59±1.96 | 3.57±2.09 | 52.43±55.49 | 51.07±49.56 | 48.96±47.10 | 54.11±50.08 | 52.51±52.76 |
| | SSIM | 0.882±0.091 | 0.881±0.094 | 0.884±0.089 | 0.836±0.117 | 0.840±0.112 | 0.945±0.035 | 0.937±0.045 | 0.933±0.050 | 0.900±0.133 | 0.901±0.128 | 0.901±0.126 | 0.869±0.174 | 0.866±0.179 |

## D.4 ABLATION STUDY: TARGET LAYER FOR EXTRACTING LATENT FEATURES

In DNNs, feature maps from different intermediate layers represent different levels of features. Specifically, early layers capture low-level and basic features including edges and textures, which are input-specific, while deep layers tend to extract high-level and more abstract features, which are model-specific (Ganeshan et al., 2019). Feature-level attacks are intuitively affected by the choice of target layer. To investigate this effect, we configure QMirage to target features extracted at different layers and report the attack results on finite-shot and noisy QNNs in Table 9. Note that HQNN is not concluded here since its total number of layers is 2.

Generally, compared to early layers, attacks on deep layers tend to converge faster and demand fewer iterations, indicating stronger sensitivity to deeper features. This is consistent with the feature attacks in DNNs (Huang et al., 2019). Nevertheless, given that the circuit scale of current QNNs is relatively small, there is no significant difference between layer-wise attacks. For larger-scale QNNs where the layer choice might introduce an obvious difference in the future, we consider disrupting latent features at each layer to avoid choosing a single layer. Moreover, layer-wise adversarial transferability between different QNNs can also be a future research direction.

## D.5 ATTACK ON LARGER-SCALE QNNS

Table 10: Attack results on larger-scale QCL (MNIST).

| | Deeper circuits | | | More qubits | |
|---|---|---|---|---|---|
| | 5 | 10 | 15 | 8 | 10 |
| ASR | 100% | 100% | 100% | 100% | 100% |
| Iterations | 42.06±26.29 | 52.71±28.86 | 56.42±33.01 | 42.06±26.29 | 60.65±33.69 |
| SSIM | 0.845±0.076 | 0.744±0.098 | 0.707±0.102 | 0.845±0.076 | 0.807±0.095 |

In both classical and quantum adversarial machine learning literature, it has been shown that increasing the capacity of classifiers can enhance the robustness to adversarial attacks (Madry et al., 2017; Lu et al., 2019). Since adversarial examples will further obscure the decision boundary of the original model, a more complicated network might be able to correctly classify these adversarial examples with a stronger fitting capability. Here, we further validate the effectiveness of QMirage against larger-scale models. For concreteness, we consider two settings to increase the capacity of QNNs: (1) increasing the circuit depth [3] by stacking more layers, and (2) using more qubits. We choose binary-classification QCL as an example. For setting (1), the circuit depth is configured as 5, 10, and 20, while keeping the qubit number at 8. For setting (2), the qubits are increased to 10 with a depth of 5. In fairness, all these models are trained and attacked in consistence with Section 5.2.2.

We find that as the circuit scales up, QMirage demands more iterations and perturbations in general, implying the stronger robustness of larger-scale QNNs to adversarial perturbations.

## D.6 COMPARISON BETWEEN IDEAL AND FINITE-SHOT MODELS

As a complement to Section 5.2, here we provide more experiments to compare the attack differences between models under ideal and realistic simulated settings. We repeat the experiments on the weight of LFD for ideal QNNs, keeping all other settings unchanged.

As shown in Figure 7, attack tendencies under the two types of QNN are distinct. Compared to realistic models, the iterations required for ideal models decrease steadily as $w$ increases, and the SSIMs do not show a sharp drop, indicating a lower sensitivity to $w$. As illustrated in Section 5.2,

---

[3] the total number of identical layers to stack

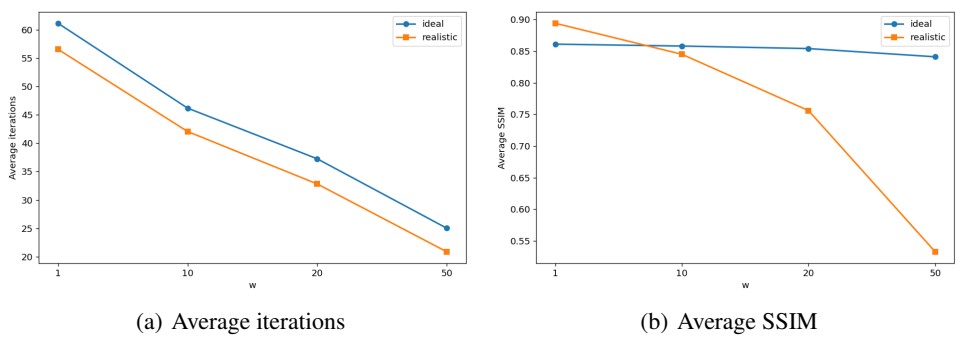

(a) Average iterations       (b) Average SSIM

Figure 7: Attack performance on ideal and realistic models (MNIST).

this phenomenon highlights the need to take into account more factors in the real quantum execution to avoid performance overestimation.

# E VISUALIZATION OF THE ADVERSARIAL EXAMPLES

Here we provide some examples of adversarial examples generated by QMirage in Figures 8 and 9.

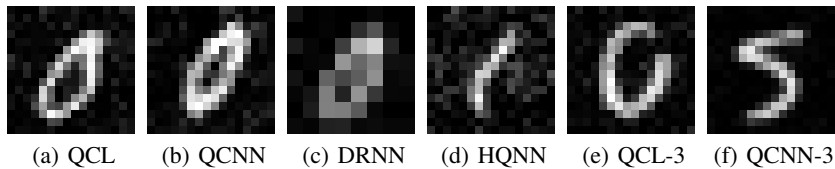

(a) QCL    (b) QCNN    (c) DRNN    (d) HQNN    (e) QCL-3    (f) QCNN-3

Figure 8: Adversarial examples generated by QMirage on realistically simulated QNNs (MNIST).

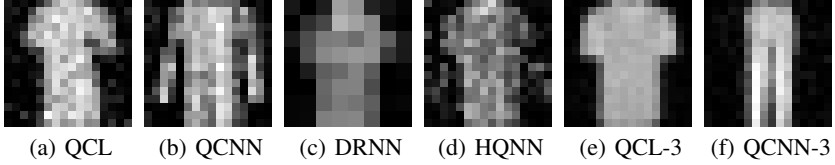

(a) QCL    (b) QCNN    (c) DRNN    (d) HQNN    (e) QCL-3    (f) QCNN-3

Figure 9: Adversarial examples generated by QMirage on realistically simulated QNNs (FashionM-NIST).

