# OpenReview forum: "Feature-Level Adversarial Attack on Quantum Neural Networks"
_ICLR.cc/2026/Conference — ICLR 2026 Conference Withdrawn Submission_

### Official Review · Reviewer_B3hM · 2025-10-14

**Soundness:** 1
**Presentation:** 3
**Contribution:** 1
**Rating:** 0
**Confidence:** 5

**Summary:**

QMirage is a feature-level adversarial attack geared toward quantum neural networks. It used quantum latent features based on superposition to optimize a constrained objective function to trigger model misbehaviors. QMirage also integrates gradient estimation with prior information to accommodate various encoding schemes to make the deployment more feasible. Instead of a white-box or black-box adversary knowledge, this work assumes a "grey-box" model, which is not realistic in a quantum setting.

**Strengths:**

The work targets an interesting topic of feature-level adversarial attack by perturbing the input image pixels in minute ways that are not detectable, leading to incorrect classification prediction by the quantum model.

The use of a classical technique for such perturbation has advantages and disadvantages. It requires fewer quantum simulations, but it is not going to be scalable beyond circuit sizes with tens of qubits.

**Weaknesses:**

The assumption that an adversary could access intermediate outputs or "activations" of a quantum neural network in a grey-box model is fundamentally unrealistic because it violates the basic principles of quantum mechanics. In a classical neural network, one can observe activations or feature maps at intermediate layers because these are deterministic numerical values stored in memory. In contrast, a quantum neural network operates on quantum states, which are superpositions of amplitudes that evolve unitarily through a sequence of gates. These states cannot be read without performing a measurement, and any measurement collapses the superposition into a single classical outcome, irreversibly destroying the quantum information. The idea that an adversary could somehow observe or extract partial information about these intermediate states assumes a kind of non-destructive readout that simply does not exist in quantum physics.

Even if one stretches the idea and imagines an adversary embedded within the system, capable of performing tomography or mid-circuit measurements, the assumption remains implausible. Quantum state tomography requires exponentially many measurements over identically prepared circuits to reconstruct even a small state, which becomes infeasible for practical systems. Moreover, any attempt to measure intermediate qubits directly interrupts the computation, collapsing those qubits and altering the output of the network. The concept of a grey box adversary, someone who can access both the model’s classical outputs and its intermediate quantum activations, is therefore a categorical mistake. It borrows the terminology of classical machine learning threat models, where partial observability is feasible, and misapplies it to a quantum system where information cannot be accessed without destruction.

Further, in my experience with real-world quantum settings such as IBM Quantum, AWS Braket, or Microsoft Azure, no interface exists for intermediate state access because the hardware itself cannot expose such data. Users and adversaries alike can only submit full circuits, execute them, and retrieve classical measurement results from the final layer. At most, an adversary might gain access to classical metadata such as gate sequences, calibration logs, timing, or error rates, but never to the internal quantum states themselves (no one can -- not even the victim). Any claim of partial observability or intermediate feature access should therefore be recognized as a purely simulation-based convenience, not a property of real quantum devices. When imported into physical threat models, this assumption completely undermines the theoretical and practical validity of the work because it presumes a form of introspection that quantum systems fundamentally forbid.

**Questions:**

No questions.

---

### Official Review · Reviewer_xyT6 · 2025-10-24

**Soundness:** 2
**Presentation:** 2
**Contribution:** 2
**Rating:** 2
**Confidence:** 4

**Summary:**

This paper proposes QMirage, a new feature-level adversarial attack framework tailored for Quantum Neural Networks (QNNs). Unlike prior approaches that rely on inaccessible gradients or assume ideal quantum states, QMirage estimates gradients using Natural Evolution Strategies (NES) and integrates latent feature distance (LFD) defined in quantum feature space. The paper further introduces a dynamic learning rate adjustment mechanism to stabilize optimization under stochastic measurement noise. Experiments on multiple QNN architectures (QCL, QCNN, DRNN, HQNN) and datasets (MNIST, FashionMNIST) show that QMirage outperforms baseline quantum and classical-inspired attacks in success rate, efficiency, and visual quality, even under noisy and finite-shot quantum conditions.

**Strengths:**

1.	The paper targets the relevant and emerging problem of adversarial robustness of QNNs.
2.	The introduction of a feature-level loss in quantum latent space (LFD) is conceptually interesting and extends an idea from classical DNNs to QNNs.
3.	Experiments include several QNN architectures and both ideal and noisy simulation conditions.
4.	The paper provides detailed algorithmic pseudocode and mathematical formulations of the proposed attack.

**Weaknesses:**

1.	The approach mostly adapts known black-box derivative-free optimization techniques (NES) and classical feature-level losses to QNNs, rather than introducing fundamentally quantum mechanisms. The contribution is incremental over previous works such as QuanTest (Shi et al., 2025) and qFGSM (Lu et al., 2019).
2.	Section 4.1 defines quantum latent features simply as measurement probability distributions, which are already used in existing QNN literature as state representations. The “feature-level” terminology appears forced, and the LFD metric (Eq. 1) is a direct application of the Jensen-Shannon distance without theoretical justification for its link to adversarial sensitivity.
3.	The threat model (Section 3) describes a grey-box setting where intermediate features are accessible, but this assumption is unrealistic for most QNN hardware, where such internal states are not measurable without collapsing quantum information. This undermines the paper’s claim of “practical applicability.”
4.	The gradient estimation using NES (Section 4.2) lacks theoretical or empirical justification for its efficiency or unbiasedness in quantum contexts. The derivations are directly lifted from Ilyas et al. (2018a,b) for classical DNNs, but the paper does not evaluate how the stochasticity of quantum measurements affects gradient quality or attack convergence.
5.	The dynamic learning rate adjustment (Section 4.3) introduces multiple hyperparameters (τₑ, ρ, τₛ, β) without sensitivity analysis. No intuition or theoretical basis is provided for the chosen thresholds or checkpoint schedule, which appear arbitrary.
6.	The experimental setup (Section 5.1) is insufficiently detailed. The circuit depths, qubit counts, and encoding specifics of QCL, QCNN, DRNN, and HQNN are only described in Appendix C.1 but not summarized in the main text. The details are too sparse to reproduce the experiments.
7.	All baselines are reimplemented using the NES adaptation. The original versions (with proper gradient access) are not reproduced for contrast, making it unclear how much of the improvement arises from NES vs. LFD.
8.	Section 5.2 claims that QMirage works under noisy and finite-shot settings, but this is tested only on simulated environments (PennyLane noise models), not on real quantum hardware. The term “practical feasibility” is therefore misleading.
9.	Measurement cost analysis (Section 5.4) is overly simplistic, since it only counts shot complexity but ignores latency and resource constraints on NISQ devices, which may make QMirage infeasible for circuits with more than a few qubits.
10.	The related work (Section 2) omits some relevant studies on QNN robustness and gradient-free quantum optimization. The discussion is heavily biased toward a few classical analogues and lacks broader context.
11.	Although Section 6 briefly mentions future directions, the paper does not connect QMirage to potential mitigation or robustness evaluation frameworks, limiting its scientific significance.
12.	The overall contribution appears minor, since the work applies an existing black-box attack framework (NES) with a handcrafted auxiliary loss to quantum settings.

**Questions:**

1.	How feasible is it to access intermediate latent features in real QNNs without collapsing quantum states, as required in the grey-box model?
2.	Has QMirage been evaluated on actual quantum hardware or only on simulators? If only simulated, how would measurement noise be expected to scale in practice?
3.	How sensitive are the results to the choice of LFD weight w, sampling population K, and variance σ?
4.	Could the NES-based gradient estimation be compared with other gradient-free quantum optimization methods (e.g., SPSA, finite-difference) to isolate its impact?
5.	What is the approximate runtime or measurement budget (in shots) per adversarial example for each model?

---

### Official Review · Reviewer_z8p1 · 2025-11-01

**Soundness:** 2
**Presentation:** 2
**Contribution:** 2
**Rating:** 2
**Confidence:** 4

**Summary:**

- The paper proposes QMirage, a feature-level adversarial attack for Quantum Neural Networks (QNNs). The core ideas are: (i) define quantum latent features as the intermediate-layer probability distribution over computational basis states; (ii) optimize a joint objective that sums a classification loss and a Jensen–Shannon distance (JSD) between benign and perturbed latent features; (iii) estimate input gradients via NES with a time-dependent prior; and (iv) adapt the step size using variance/monotonicity checks. Experiments on MNIST/FashionMNIST with several small QNNs (QCL, QCNN, DRNN, HQNN) claim higher ASR, fewer iterations, and good SSIM vs. qFGSM/qBIM/CW/QuanTest; additional results with finite-shot and noisy simulators suggest robustness.

**Strengths:**

- Adversarial evaluation for QNNs is under-explored; bringing feature-space thinking from DNNs into QNNs is interesting.

**Weaknesses:**

- The threat model appears unrealistic, as it assumes that the attacker has access to intermediate quantum features (e.g., layer-wise probability distributions or activations) while lacking access to model parameters or gradients. This constitutes an unusually strong and impractical assumption for quantum neural networks, given that mid-circuit measurements on current hardware are typically destructive and reconstructing such intermediate states would require costly procedures such as tomography or duplicated subcircuits. Without a clear explanation of how this access is achieved in practice, the proposed setting lacks credibility as a realistic adversarial scenario.
- The feature-space attack concept closely mirrors classical Intermediate-Level Attack (ILA) or feature-disruption methods, with the main quantum-specific contribution being the definition of the “feature” as a probability vector. Similarly, the use of NES with time-dependent priors follows the formulation by Ilyas et al.; while its application to QNN inputs is reasonable and useful, the underlying technique itself is not new.
- The evaluation is insufficiently comprehensive. The baseline set omits decision-based and score-only black-box attacks (e.g., HSJA, Square Attack and its variants), which are particularly relevant under practical constraints where gradients or internal model details are inaccessible. Including such baselines would provide a fairer and more realistic comparison.
- No hyperparameter sensitivity study is provided. The paper lacks an analysis of how key parameters (e.g., the weight of the feature loss, number of samples, noise scale, or step-size factors) affect the attack’s stability and effectiveness, making it difficult to assess the robustness and reproducibility of the proposed method.
- the paper does not examine whether adversarial examples generated on one QNN transfer to other architectures or configurations, which limits understanding of the attack’s generality.

**Questions:**

Please check weaknesses.

---

### Note · Authors · 2025-11-15

I have read and agree with the venue's withdrawal policy on behalf of myself and my co-authors.